# Alginate/Pectin Film Containing Extracts Isolated from Cranberry Pomace and Grape Seeds for the Preservation of Herring

**DOI:** 10.3390/foods12081678

**Published:** 2023-04-18

**Authors:** Gabrielė Urbonavičiūtė, Gintarė Dyglė, Darius Černauskas, Aušra Šipailienė, Petras Rimantas Venskutonis, Daiva Leskauskaitė

**Affiliations:** 1Department of Food Science and Technology, Kaunas University of Technology, Radvilenu pl 19, LT-50254 Kaunas, Lithuaniaausra.sipailiene@ktu.lt (A.Š.);; 2Food Institute, Kaunas University of Technology, Radvilenu pl 19, LT-50254 Kaunas, Lithuania

**Keywords:** alginate/pectin film, cranberry pomace extract, grape seeds extract, herring, biogenic amines

## Abstract

Alginate/pectin films supplemented with extracts from cranberry pomace (CE) or grape seeds (GE) were developed and applied to herring fillets that were stored for 18 days at 4 °C. Herring coated with films containing GE and CE inhibited the growth of *Listeria monocytogenes* and *Pseudomonas aeruginosa* during the storage period, whereas pure alginate/pectin films did not show an antimicrobial effect against the tested pathogens. The application of alginate/pectin films with CE and GE minimised pH changes and inhibited total volatile basic nitrogen (TVN) and the formation of thiobarbituric acid-reactive substances (TBARS) in the herring fillets. The coating of herring fillets with films with CE or GE resulted in three- and six-fold lower histamine formation and one-and-a-half- and two-fold lower cadaverine formation, respectively, when compared to unwrapped herring samples after 18 days of storage. The incorporation of 5% extracts isolated from cranberry pomace or grape seeds into the alginate/pectin film hindered herring spoilage due to the antimicrobial and antioxidant activity of the extracts.

## 1. Introduction

Since fish is one of the most perishable foods, the search for a means to preserve it remains a relevant topic among food researchers and technologists. The main reason for the spoilage of fish and fish products is direct microbial propagation or the emergence of bacterial enzymes and their metabolism of fish nutrients such as biogenic amines (BAs) [1]. BAs such as histamine (HI), putrescine (PUT), cadaverine (CAD), spermidine (SPD), spermine (SPM), tyramine (TY), phenylethylamine (PHY) and tryptamine (TR) are formed by the decarboxylation of free amino acids as a result of metabolic processes in microorganisms [2]. Delaying the spoilage of fish involves inactivating pathogens and preventing BA formation by inhibiting bacterial growth or decarboxylase enzyme activity, and several such methods have recently been proposed [3]. Edible films containing natural products with preservative properties are very promising for a variety of reasons. Firstly, they are composed of food-grade biopolymers (proteins or polysaccharides) that are biodegradable [4,5]. Secondly, edible films can act as carriers of functional ingredients (e.g., antimicrobials, antioxidants) from natural sources [6]. Applied directly to fish fillets by forming a thin layer of biopolymer with incorporated functional ingredients, edible films can provide considerable protection against microbial growth and metabolism that can result in the formation of toxic metabolites (e.g., BAs) [7].

The inclusion of various plant extracts in edible films or coatings proved to be an effective strategy for improving fish quality and increasing its shelf life. Products packaged in the alginate edible film containing 15% *Aloe vera* extract showed significantly higher lipid stability and microbial quality [8]. Farsanipour et al. (2020) used chitosan in combination with whey protein isolate and tarragon essential oil for the preparation of edible films and applied them to *Scomberoides commersonnianus* fillets [9]. The authors reported the strong activity of the prepared edible films against bacteria alongside improved antioxidant activity. The preservation of rainbow trout fillets was also improved by coating them with a carboxymethylcellulose film incorporated with *Zataria multiflora* Boiss essential oil, grape seed extract (0.5–1%) and combinations thereof [10]. The inhibition of microbial growth was also observed in a chitosan coating with either grape seed extract or tea polyphenols during storage to preserve red drum fillets [11]. Another study indicated that the edible films of sodium alginate-carboxymethylcellulose matrix incorporated with *Ziziphora clinopodioides* essential oil, apple peel extract or zinc oxide nanoparticles increased the shelf life (microbial, chemical and sensory properties) of silver carp fillet and inhibited the growth of *Listeria monocytogenes* during the refrigerated storage of fish over a period of 2 weeks [12]. Similarly, a film composed of sodium alginate and tea polyphenols applied on bream fillets enhanced the shelf life of fish fillets from 15 to 21 days [13].

In the case of BA formation in fish, the use of various plant extracts with antimicrobial activity has shown promising inhibitory effects [14]. Extracts prepared from safflower (*Carthamus tinctorius*) and bitter melon (*Momordica charantia*) demonstrated bactericidal effects on some fish spoilage (*Acinetobacter lwoffii*, *Pseudomonas oryzihabitans*, *Enterobacter cloacae*, *Shigella* spp., *Morganella psychrotolerans* and *Photobacterium phosphoreum*) and food-borne pathogens (*Staphylococcus aureus*, *Klebsiella pneumoniae* and *Salmonella* Paratyphi A). They also reduced the number of BAs, such as HI, PUT, CAD, SPD, SPM, TY and PHY [15]. Furthermore, the treatment of vacuum-packed sardine fillets with mint (*Mentha spicata*) and sagebrush (*Artemisia campestris*) extracts restricted the production of BAs and reduced the presence of HI, TY and CAD [16].

Recently, several investigations have focused on edible active packaging technologies that can reduce and control BA levels in fish. However, published research results are contradictory. For example, active double-layered furcellaran/gelatin hydrolysate films containing Ala-Tyr peptide were developed and used on Atlantic mackerel stored at −18 °C for 4 months [17]. In general, the use of films did not inhibit the formation of BAs in these mackerel samples. Moreover, in the case of TY and SPD, the use of the film increased their formation. Meanwhile, a study conducted by Hao et al. (2017) showed that in abalone (*Haliotis discus hannai*) treated with a sodium alginate coating and bamboo leaf extract or rosemary extract stored under chilled conditions, PUT and CAD were the main BAs [18]. Moreover, the authors noted that coating abalone with rosemary extract greatly inhibited BA formation. Both the total BAs and specific BAs were far below the recommended limits.

Although many studies have reported the effects of edible films containing plant extracts rich in phenolic compounds on fish quality and the inhibition of microbial growth, few reports have investigated the influence of edible films combined with plant extracts on BA formation. Therefore, our study aimed to incorporate phenol-rich berry extracts into alginate/pectin films to develop edible packaging for the active preservation of fish products. The antimicrobial effectiveness of cranberry pomace extract and grape seed extract incorporated in alginate/pectin films on the growth of food-borne pathogens (*L. monocytogenes* and *P. aeruginosa*) and BA formation was evaluated in herring fillets. Finally, the quality properties (pH, TBARS, volatile nitrogenous base formation) of herring fillets with edible films applied during storage were estimated. To the best of our knowledge, this is the first study on the synergy of cranberry pomace extract or grape seed extract and alginate/pectin films in the preservation of herring.

## 2. Materials and Methods

### 2.1. Materials and Chemicals

Skinned herring (*Clupea harengus*) fillets with an average weight of 120 ± 10 g were supplied by a local retailer (Orkos, Kaunas, Lithuania).

Food-grade alginic acid sodium salt from brown algae (low viscosity) (Sigma-Aldrich^®^, Darmstadt, Germany) and pectin from citrus peel (Sigma-Aldrich^®^, Darmstadt, Germany) were provided by Labochema (Vilnius, Lithuania). Calcium chloride for the cross-linking reaction and glycerol for improving film plasticity were obtained from Eurochemicals (Vilnius, Lithuania).

Biogenic amines, namely, putrescine (PUT, C_4_H_12_N_2_), cadaverine (CAD, C_5_H_14_N_2_), histamine (HI, C_5_H_9_N_3_), tyramine (TY, C_8_H_11_NO) and spermine (SPD, C_10_H_26_N_4_)) standards, were purchased from Sigma-Aldrich (Darmstadt, Germany).

The frozen cranberries were donated by the Fudo Company (Kaunas, Lithuania). Thawed berries were pressed in a Philips HR1880/01 juicer. The pomace was air dried at 35 °C (final moisture content: 5.83%) and ground in a centrifugal mill (Retsch ZM200, Haan, Germany) using a 0.2 mm sieve. The lipophilic fraction of dried cranberry pomace was isolated in a pilot-scale supercritical CO_2_ extractor (Applied Separation, Allentown, PA, USA) as previously described [19]. The defatted residue was extracted with agricultural-origin ethanol (Stumbras, Kaunas, Lithuania) in a pilot-scale Soxhlet-type extractor. After extraction, the ethanol was removed in a rotary vacuum evaporator and stored at 4 °C until used. The extract (CE) contained 111.29 ± 0.24 mg/g of polyphenolics (expressed in gallic acid equivalents (GAE)) and 333.1 ± 7.0 mg/100g of procyanidins (determined spectrophotometrically).

Grape seed extract (GE) was purchased from DRT the Best of Nature (Vielle-Saint-Girons, France). The GE contained 626.32 ± 12.96 mg GAE/g of polyphenolics and 1161.00 ± 66.47 mg/100 g procyanidins (determined spectrophotometrically). Overall, 19.6% of procyanidins were monomers, and 40.8% were dimers and trimers. Consequently, the CE and GE can be considered rich in phenols.

All culture media for microbial analysis were supplied by Biometrija (Kaunas, Lithuania): Plate Count Agar (REF 310040, Liofilchem, Italy), Pseudomonas Agar Base (CM559 Oxoid), CFC Supplement (SR103), Agar Listeria Ottaviani Agosti (REF 4016052 Biolife), ALOA Enrichment Supplement (REF 423501) and ALOA Selective Supplement (REF 423501).

Analytical-grade chemicals were obtained from various suppliers: perchloric acid (HClO_4_, Chempur, Germany); 1,7-diaminoheptane (C_7_H_18_N_2_, Sigma-Aldrich, Germany); sodium hydroxide (NaOH, Eurochemicals, Lithuania); sodium bicarbonate (NaHCO_3_, Lachema, Czech Republic); dansyl chloride (5-dimethylaminonaphtalene-1-sulfonyl chloride, Sigma-Aldrich, Germany); ammonia (25%, NH_3_, Chempur, Germany); ammonium acetate (0,1 mol/l, CH_3_COONH_4_, Reachem, Slovakia); acetonitrile (Carlo Erba, France); HCl (Stanlab, Poland).

### 2.2. Test Microorganisms

*Staphylococcus aureus* ATCC 25923, *Listeria monocytogenes* ATCC 19117, *Pseudomonas aeruginosa* ATCC 27853, *Salmonella* Typhimurium ATCC 14028, *Escherichia coli* ATCC 8739 and *Escherichia coli* NCTC 12900 were provided by Biometrija (Kaunas, Lithuania). For the preparation of bacterial suspension, the microorganisms were grown overnight on a slant Plate Count Agar (REF 310040, Liofilchem, Italy) at 37 °C and suspended in sterile saline (0.85% NaCl) to achieve a cell concentration of 10^5^–10^6^ colony-forming units per millilitre (cfu/ml), corresponding to 0.5 McFarland.

### 2.3. Preparation of Alginate/Pectin Films with Extracts

First, a mixture was made by mixing alginate and pectin powder in a ratio of 7:3. Then, 3 g of the alginate/pectin mixture was dissolved in 100 mL of 1% (*w*/*w*) glycerol. Solubilisation was carried out at 30 ± 1 °C for 1 h with continuous stirring on a magnetic stirrer/hot plate. After adding 5 g of CE or GE to the solution, it was further stirred at 30 ± 1 °C for 30 min. For film formation, 10 g of each film-forming solution was poured into Petri dishes (90 mm diameter), which were placed on flat, level trays and kept at 35 ± 1 °C for 48 h (KB8182 in a cooling incubator, Termaks, Bergen, Norway). The film without extract was prepared and referred to as the control with film without extracts. The films were cross-linked by spraying with a 5% (*w*/*w*) calcium chloride solution. The cross-linked films were peeled from the Petri dishes and stored in desiccators at 65% RH for further analysis. Films without extract were used as controls.

### 2.4. Evaluation of Antimicrobial Properties of Film via the Agar Diffusion Method

The antibacterial properties of the film were evaluated by agar diffusion assay [20] with slight modification. First, 1 mL of tested *S. aureus* ATCC2 5923, *L. monocytogenes* ATCC 19117, *S.* Typhimurium ATCC 14028, *E. coli* ATCC 8739 and *E. coli* NTCT 12900 bacterial suspensions were prepared separately for each bacterial strain (as previously described) and mixed with 100 mL of the Plate Count Agar (REF 310040, Liofilchem, Italy). The obtained mixture of media and bacterial suspension was poured into Petri dishes (90 mm diameter) at 12 mL each and allowed to solidify surface.

The pieces of the film (10 × 10 mm) were cut and placed onto the agar surface. For the antimicrobial assessment of extracts, their 5% (*w*/*v*) water solutions were prepared and poured into the agar wells (diameter 8 mm). The Petri dishes were incubated in a KB8182 incubator (Termaks) at 37 °C for 24 h. After incubation, the inhibitory zone around the coating pieces and/or the contact area of the film with the agar surface was measured with a Vernier caliper. Results were expressed (in mm) as average zone areas.

### 2.5. Preparation of Herring Fillets

Herring fillets were aseptically cut into 10.00 ± 0.05 g samples. Then, the samples were fully coated with prepared film. The coated samples were placed in Petri dishes and stored at 4 ± 1 °C for 18 days. Samples were randomly collected after storage for 1, 2, 4, 6, 12 and 18 days for chemical, microbiological and biogenic amine (BA) analysis. For the determination of tested microbial counts in herring, the samples were spread with 10-μL bacterial suspensions of *P. aeruginosa* ATCC 27853 or *L. monocytogenes* ATCC 19117 (prepared as previously described) to achieve a level of 4.0 lg cfu/g. All samples were analysed after the removal of the film.

### 2.6. Determination of Tested Microorganism Counts in Herring

Total bacterial counts in coated herring samples were determined after 0, 1, 2, 4, 6, 12 and 18 days of storage. After removal of the film at each sampling interval, 10 g fillet portions were prepared in ten-fold dilutions. Plates inoculated via the streak plate method were incubated aerobically at 37 °C for 48–72 h. *P. aeruginosa* and *L. monocytogenes* bacteria counts were determined on Pseudomonas Agar Base (CM559 Oxoid) with CFC Supplement (SR103) and Agar Listeria Ottaviani Agosti (REF 4016052 Biolife) with ALOA Enrichment Supplement (REF 423501), respectively. The results were expressed as lg cfu/g.

### 2.7. Determination of Biogenic Amines (BAs)

BAs were quantitatively identified by high-performance liquid chromatography (HPLC). For extraction, 5 g of sample was homogenised with 20 mL of perchloric acid solution (0.4 mol/L) in a 50 mL screw-cap tube, and 250 μL of internal standard 1,7-diaminoheptane stock solution (1 mg/mL) was added to achieve a 1 μg/mL concentration in the injection volume. The mixture was centrifuged (MPW-260RH, MPW Med. Instruments, Poland) at 4000 rpm, and the supernatant was rinsed into a 25 mL bottle through Whatman no. 1 filter paper (180 μm thickness and 11 μm particle retention rating at 98% efficiency). Filtrate was adjusted to 25 mL with a perchloric acid solution (0.4 mol/L).

For the derivatisation, 500 μL of sample extract was made alkaline by adding 100 μL of sodium hydroxide solution (2 mol/L). The sample was then buffered by adding 150 μL of saturated sodium bicarbonate. Then, 1 mL of dansyl chloride (5-dimethylaminonaphtalene-1-sulfonyl chloride) solution (10 mg/mL) was added and mixed thoroughly using a shaker-mixer (IKA mini Shaker TS1, Germany). The reaction mixture was then transferred to a 40 °C incubator for 45 min. After incubation, it was cooled to room temperature for 10 min, and residual dansyl chloride was removed by adding 50 μL of ammonia (25%). The mixture was then mixed with a shaker-mixer. After 30 min, it was adjusted to 5 mL with an ammonium acetate (0.1 mol/L):acetonitrile mixture (1:1, *v*/*v*) and mixed well with a shaker-mixer. The mixture was then filtered through a 0.20 μm nylon filter (UptiDisc, Interchim, France), and the solution was injected into an analytical column.

A Shimadzu Prominence LC20AD (Shimadzu, Japan) coupled to a UV detector SPD/20 A chromatographic system was utilized with a LabSolution (Shimadzu, Japan) integrator using a Hydrosphere C18 (5 µm, 12 nm), 150 × 4.6 I.D. column and YMC pre-column ProC18 (10 × 3.0 mml.D., S-3 μm, 12 nm) (YMC Co., Ltd., Japan). LC mobile phase A: ammonium acetate (0.1 mol/L), phase B acetonitrile. Operating conditions: flow rate 0.9 mL/min; injection volume 20 μL; column temperature 40 °C; peaks were detected at 254 nm; gradient 50% B to 90% B in 19 min; run time 20 min; post-run before next run, 50% B, 8 min.

Five BAs were quantified: PUT, CAD, HI, TY and SPD. Stock solutions (1 mg/mL) for each amine were prepared in 0.1 M HCl and stored at 4 ± 1 °C. For amine identification, standard solutions of individual BAs were chromatographed separately and mixed to determine the retention times and responses of each. Standard curves with correlation coefficients for stock solutions were obtained via the external standard method. All results were expressed in mg/kg.

### 2.8. Determination of the Total Volatile Basic Nitrogen (TVN)

TVN was determined according to Commission Regulation (EC) No. 2074/2005 [21]. Briefly, 10 ± 0.1 g of ground herring sample was mixed with 90.0 mL perchloric acid solution (6 g/100 mL) and filtered through No. 2 Whatman filter paper. Then, 50.0 mL of the obtained extract was subjected to steam distillation in a 2100 Kjeltec Distillation Unit (FOSS Tecator AB) following alkalinisation with 0.2 M NaOH. Then, 100 mL of distillate was collected in 100 mL of 3% (*w*/*v*) boric acid solution containing Tashiro’s indicator. The TVN contained in the distillate solution was determined by titration with 0.01 M HCl, and its concentration was calculated via the following formula:TVN (mg/100 g sample) = (V_1_ − V_0_) × 0.14 × 2 × 100/W, (1)
where V_0_ is the volume of hydrochloric acid used in blank titration; V_1_ is the volume of hydrochloric acid used in sample titration; and W is the weight of fish sample in grams.

### 2.9. pH Measurements

pH was measured directly in fish samples using an XS instrument pH7 digital pH meter (EU, P.R.C.) with a penetrating probe.

### 2.10. Determination of Thiobarbituric Acid-Reactive Substances (TBARS)

A TBARS assay was performed to evaluate lipid oxidation as described by Tarladgis et al. (1960) [22] with some modification. First, 10 g of homogenised sample were mixed with 98 mL of distilled water. Then, 1.25 mL of 0.4 N HCl were added to 50 mL of the prepared mixture. The mixture was then heated with steam distillation. Subsequently, 5 mL of distillate and 5 mL of thiobarbituric reactive reagent containing 0.02 M TBA in 90% glacial acetic acid were transferred into a glass tube. The tube was then placed into a boiling water bath for 1 h. After cooling, the absorbance of the pink solution was measured at 532 nm using a Biochrom spectrophotometer (Holliston, MA, USA). The constant 7.8 was used to calculate the TBARS number. The TBARS value was expressed as mg malonaldehyde equiv. (MDA)/kg sample.

### 2.11. Statistical Analysis

Data are expressed as means ± SD. An analysis of variance (ANOVA) followed by Fisher’s LSD test was performed using Statistica 8.0 software. A *p*-value of <0.05 was considered statistically significant. All experiments and analyses were run in triplicate.

## 3. Results and Discussion

### 3.1. Antimicrobial Properties of Films with CE and GE

To gain insights into the antimicrobial properties of CE, GE and alginate/pectin films with extracts, the inhibition zone diameters against *S. aureus* ATCC25923, *L. monocytogenes* ATCC19117, *S.* Typhimurium ATCC14028, *E. coli* ATCC8739 and *E. coli* NTCT12900 were determined (Figure 1). It should be noted that the amount of incorporated extracts in the 10 × 10 mm (1 cm^2^) piece of film, which was used in this assay, was approximately three-fold higher than that directly added to the well with the extract solution, 7.9 mg vs. 2.6 mg.

This preliminary evaluation indicated that GE both in pure form and incorporated into the film in most cases was a stronger antimicrobial agent than CE, except for GE extract’s effect on *L. monocytogenes*, when its activity was not significantly different from the CE. The antimicrobial activity of films with CE was not significantly (*p* < 0.05) different compared with pure extract against the tested bacteria, except for *E. coli* NTC12900, when the film with CE demonstrated lower activity than the extract. The results showed that the films with CE produced inhibition zones in the range of 10.0–14.0 mm. Such zones of inhibition can be considered small. However, the antimicrobial activity of the film with CE was detected against all microorganisms tested, and a wide impact is a definite advantage of the film supplemented with CE. The antimicrobial activity of different cranberry extracts alone against *S. aureus* and *E. coli*, *S. enteritidis* and *L. monocytogenes* has been shown by several studies [23,24,25]. It was concluded that both a low pH and the concentration of bioactive phenolics are responsible for the antimicrobial properties of cranberries [26]. However, direct comparison of our results with those previously published is very difficult because of different extract preparation procedures and the doses applied. For instance, Côté et al. (2011) tested full cranberry juice and extracts [23], and Laplante et al. (2012) used proanthocyanidin standardized cranberry extracts [24]. The latter study demonstrated that antimicrobial effects were strongly dependent on the proanthocyanidins concentration; the minimal inhibitory concentration (MIC) of the extracts with 210 mg/g in most cases was >30-fold lower than that of the extracts with 8 mg/g [24]. For comparison, the concentration of proanthocyanidins in CE applied in our study was only 3.33 mg/g. Some other studies inoculated whole cranberry extracts in ground beef [25], and cranberry pomace extracts were tested in the commonly used meat fermentation starter cultures [26]. Severo et al. (2021) demonstrated that chitosan films with the incorporated whole CEs inhibited *E. coli* and *S. aureus* biofilm formation, which was explained by the presence of phenolic compounds [27]. Similarly, our previous studies on whey protein-chitosan film incorporating cranberry juice showed their antimicrobial activities against *S*. Typhimurium, *L. sakei*, *L. plantarum*, *S. agona* and *C. jejuni* [28].

Our study showed significantly (*p* < 0.05) higher sensitivity of the tested microorganisms against the films supplemented with GE in comparison to the films with CE. It is interesting to note that the applied amounts of CE and GE in most cases gave similar inhibition zones (the differences were not significant) for the extract solution and the film. This may be explained by the fast impact of inhibiting compounds, which are rapidly diffusing into the bacteria culture after dosing into the well, whereas their release from the cross-linked films proceeds more slowly. From this point of view, approximately three-fold higher amounts of extracts in the films were a quite reasonable selection for this assay. However, the antimicrobial effects were also dependent on the bacteria species. Thus, among tested food-borne pathogens, the smallest inhibition zone was obtained for *E. coli* NTCT, with an average value of 17.7 ± 1.5 mm, whereas for *S.* Typhimurium, the inhibition zone of the film with GE was remarkably larger, 29.3 ± 6.1 mm. Again, the good antimicrobial activity of the film with GE can be attributed to the phenolic compounds present in GE [29]. Our study used GE containing a three-and-a-half-fold higher concentration of proanthocyanidins than CE. For instance, Sogut and Seydim (2018) demonstrated that GE-incorporated chitosan films inhibited *E. coli*, *L. monocytogenes*, *S. aureus* and *P. aeruginosa* more efficiently than chitosan films alone, and the effect of 5% GE (the concentration used in our study) on the count of coliform bacteria was significantly lower that that of 10 and 15% [30].

Our results for the antimicrobial properties of alginate/pectin films supplemented with CE and GE suggest that they can be used in food systems to control food-borne pathogens. Therefore, further antimicrobial studies were performed by applying films on herring that had previously been infected with pathogenic bacteria.

### 3.2. Herring Preservation by Applying Alginate/Pectin Films with CE or GE

#### 3.2.1. Viability of *L. monocytogenes* and *P. aeruginosa*

In general, the viable cell count method is used to assess the antimicrobial effectiveness of edible films for coated or wrapped fish products. Figure 2 presents the viability results of *L. monocytogenes* and *P. aeruginosa* in herring wrapped in alginate/pectin films supplemented with CE or GE during storage for 18 days. *L. monocytogenes* can be embedded into fish during handling or storage processes, and *P. aeruginosa* is a normal part of the fish microbiota. The tested pathogens showed different sensibilities to films with CE and GE. During storage, the *L. monocytogenes* content in control samples (without film and wrapped in a film without extract) increased from 3.87 to 7.52 lg CFU/g and from 3.75 to 6.69 lg CFU/g, respectively (Figure 2A). Wrapping herring samples in the film with CE resulted in more moderate growth of the *L. monocytogenes* population: during 12 days of storage, it increased from 3.75 to 6.07 lg CFU/g and remained unchanged during subsequent storage. The herring treatment with a film containing GE had a significant (*p* < 0.05) influence on the growth of *L. monocytogenes* during storage: after 18 days, the population of *L. monocytogenes* remained at the same level observed at the beginning of storage. The applied films in the herring samples showed stronger inhibitory effects on gram-positive *L. monocytogenes* bacteria as compared to gram-negative *P. aeruginosa* bacteria.

Notably, weak growth inhibition during storage was observed for *P. aeruginosa* in herring samples wrapped in films containing CE and GE with 0.31 and 0.23 lg reductions, respectively (Figure 2B). The antibacterial effect of polysaccharide films with incorporated anthocyanins is related to the interaction between them and bacterial membranes. Several mechanisms have been suggested to explain the antibacterial action of anthocyanins, such as differences in cell wall structure, cell physiology, metabolism, destabilisation or permeability of the cytoplasmic membrane [31,32]. In control samples (without film and wrapped in film without extract), the *P. aeruginosa* content increased from 3.75 to 4.36 lg CFU/g after 18 days of storage. Our results are in accordance with Nešić et al. (2017), who indicated that pure alginate/pectin films did not show any antimicrobial effect against tested *S. aureus*, *E. coli* and *C. albicans* pathogens [33]. However, alginate/pectin films were demonstrated to be good vehicles for antimicrobial substances. The major advantage of composite alginate/pectin films with natamycin addition is their low diffusion coefficient [34]. De’Nobili et al. (2015) indicated that alginate and pectin composite film supplemented with ascorbic acid has great potential to be used in antimicrobial packaging to inhibit food spoilage [35]. Undoubtedly, the composition of polyphenols with antimicrobial properties, as well as the amount added to a film, have a significant effect on the antimicrobial properties of films [36]. In our study, GE contained considerably more polyphenols in comparison to CE (626.32 ± 12.96 mg/g and 111.29 ± 0.24 mg/g, respectively). Furthermore, the presence of a high amount of procyanidins in GE—with dimers and trimers predominating (40.8%)—may have also led to better inhibition of pathogenic microorganisms determined for the film with GE. Recent studies on novel food packaging materials containing procyanidins demonstrated outstanding antimicrobial activity [37]. Chitosan film with procyanidins was also applied in salmon muscle perseveration and showed the potential ability to prevent microorganism contamination and texture deterioration for 10 days.

#### 3.2.2. Biogenic Amines

The effect of alginate/pectin films alone—and with CE or GE—on the BA production in herring fillets during storage was investigated. The results of HI, CAD, PUT and TY contents during the storage of unwrapped herring samples and those wrapped in different films are presented in Table 1.

HI remained at low levels in all samples at the early stages of herring storage (6 days). After 18 days of storage, HI accumulation in unwrapped samples was generally higher (358.80 ± 10.32 mg/kg) than in samples coated with films. Moreover, films with CE or GE inhibited the formation of HI more efficiently in comparison to film alone. At the end of the storage period, the HI content in the samples coated with films with CE and GE was 19.52 ± 2.86 mg/kg and 31.11 ± 2.96 mg/kg, respectively. A significant (*p* < 0.05) increase in CAD was observed in all herring fillet samples during storage for 18 days. The unwrapped sample showed a higher CAD level (13.02–19.54 mg/kg) at the beginning of storage, which greatly increased up to 139.03 ± 6.32 mg/kg and 438.99 ± 9.48 mg/kg after 12 and 18 days of storage, respectively. In the film-coated samples, the CAD level exceeded 5 mg/kg only after 6 days of storage. Although the amount of CAD in the film-coated samples also increased during further storage, its amount at the end of storage was approximately two-fold lower when compared to the control sample.

Furthermore, the coating of herring fillets with alginate/pectin films resulted in decreased PUT formation during 12 days of storage. After 18 days of storage, PUT content in the unwrapped samples was lower in comparison to film-coated samples. However, these differences were statistically insignificant and relatively low. Moreover, levels of TY higher than 5 mg/kg were only detected in all samples after 12 days of storage. After 18 days of storage, the TY content in the unwrapped samples was 25.9 ± 4.24 mg/kg. In samples coated with film alone, the TY content was 49.43 ± 5.98 mg/kg. Additionally, in samples coated with film with CE or GE, the TY content was 28.01 ± 2.84 mg/kg and 6.07 ± 0.18 mg/kg, respectively. Thus, only herring fillets with a film coating containing GE inhibited the formation of TY during storage.

In our study, high levels of all analysed BAs in the unwrapped herring fillets were observed at 18 days of storage: 354.80 ± 10.32 mg of HI, 438.99 ± 9.48 mg of CAD, 76.81 ± 5.81 mg of PU and 25.9 ± 4.24 mg of TY in 1 kg of muscle. Our results are similar to those reported by Özogul et al. (2002), who found that the storage of herring fillets for 16 days without ice resulted in the accumulation of HI, CAD, PU and TY levels of 369.4, 329.9, 74.2 and 0 mg/kg, respectively [38]. Coating herring fillets with alginate/pectin film supplemented with CE or GE resulted in approximately three- and six-fold lower HI formation and about one-and-a-half- and two-fold lower CAD formation when compared to unwrapped herring samples after 18 days of storage. The effectiveness of extracts on reducing PUT accumulation in herring fillets was not observed. TY production was only considerably suppressed in the presence of films with GE. Previous research has shown that BA formation by fish spoilage bacteria in the presence of various plant extracts was dependent on the bacterial strain, extract type and dose [39]. Although *Morganella morganii*, *Klebsiella pneumoniae* and *Hafnia alvei* are reported as the strongest producers of BAs [40], Gram-positive bacteria (*S. aureus*, *E. faecalis* and *L. monocytogenes*) were able to decarboxylate more than one amino acid and produce HI, CAD and other amines [41]. Gram-negative bacteria *(E. coli*, *K. pneumoniae*, *A. hydrophila* and *P. aeruginosa)* have also been identified as BA-producing bacteria [15]. In our study, the antimicrobial effect of films containing polyphenol-rich CE and GE against *S. aureus* ATCC25923, *L. monocytogenes* ATCC19117, *S.* Typhimurium ATCC14028, *E. coli* ATCC 8739 and *E. coli* NTCT 12900 was observed. Therefore, the identified negative influence of extracts on the formation of BAs can be related to the recorded antibacterial properties against bacteria that can be BA producers.

Similarly, HI, CAD and PUT accumulation in vacuum-packed sardine fillets was suppressed by sage tea (*Salvia officinalis*) and rosemary (*Rosmarinus officinalis*) extracts [42]. Others have observed the high antimicrobial activity of garlic extract against *Bacillus licheniformis* strains and the greatest inhibitory effect on BA content in Korean salted and fermented anchovy [43]. Additionally, Özyurt et al. (2012) suggested that the icing of sardine (*Sardinella aurita*) along with rosemary extract maintained BA content at low levels, especially HI and PUT [44]. We found no data regarding the effect of CE on BA formation. Notably, GE was proven to decrease the formation of BAs in bacon: TY (30.7%), CAD (37.1%), PUT (29.4%) and HI (73.6%) [45].

#### 3.2.3. Monitoring pH during Storage

The observed pH value changes of herring fillets wrapped in alginate/pectin films with and without CE or GE are presented in Figure 3.

At the beginning of storage, the pH of all samples was not different (pH = 6.56). During 12 days of storage, the pH values of herring fillets wrapped in films with CE or GE remained constant or slightly decreased. After 18 days of storage, a moderate increase in the pH value (approximately pH 7.0) was recorded for these samples. A different change in pH was observed during the storage of control samples (unwrapped or wrapped in alginate/pectin film without extract). The pH of unwrapped herring fillets continuously increased from 6.56 to 7.05 during 6 days of storage and then drastically increased up to 8.17 after 18 days of storage. Similarly, the pH values of herring fillets coated in film without extract increased to their highest value, 8.06, after 18 days of storage. The increase in pH could be due to autolytic processes such as endogenous enzymes and microbiological enzymatic action causing protein degradation and the accumulation of nitrogen compounds such as primary, secondary and tertiary amines. At the end of storage, these processes were more intensive in the control samples in comparison to the herring samples coated with films with extract, as recorded by TVN value changes (Figure 4). The results indicate that alginate/pectin films with CE and GE significantly (*p* < 0.05) minimised pH changes and hindered herring spoilage due to the antimicrobial and antioxidant activity of polyphenol-rich extracts. Similar results were previously observed during the storage of *Scomberoides commersonnianus* coated with chitosan and chitosan-whey protein films with tarragon essential oil [9] and common carp samples treated with *Carum copticum* and lactic acid [46].

#### 3.2.4. Total Volatile Basic Nitrogen (TVN)

At the beginning of storage, the TVN value was 3.6 mg/100 g in all samples (Figure 4). During storage, TVN values increased by approximately 10 times and reached 32.26 mg/100 g and 34.00 mg/100 g in control samples without film and coated with film without extract, respectively. However, in the herring samples wrapped in films with CE and GE, significantly (*p* < 0.05) lower values of TVN were detected at the end of storage when compared to control samples. After 18 days of storage, the TVN values were 19.29 mg/100 g in the samples coated with film with CE and 14.40 mg/100 g in the samples coated with film with GE. Furthermore, in all control samples, the TVN values exceeded 10 mg/100 g—even after 12 days of storage. In the samples coated with films containing extracts at 12 days of storage, the TVN values were between 5.57 and 5.61 mg/100 g. From the obtained data, it can be stated that alginate/pectin film alone does not protect herring from the formation of volatile nitrogen bases, and the incorporation of CE or GE into the films has almost the same effect on reducing the degradation of nitrogenous compounds in herring fillets. In our studies, TVN values correlated well with the pH data.

The TVN parameter identifies primary, secondary and tertiary amines and is recognised as an indicator of muscle tissue deterioration [47]. In light of the recommendations of the European Commission (CEC) (1995) [48] and various scientists [49,50], 25–35 mg of N per 100 g is an upper acceptability limit for spoilage initiation in fresh fish. In our study, only those herring samples that were wrapped in film with CE or GE demonstrated TVN values below this limit of acceptability after 18 days of storage. Similar results were reported by Günlü and Koyun (2013) in sea bass (*Dicentrarchus labrax*) fillets wrapped with chitosan-based edible film during cold storage at 4 °C [51]. According to Farsanipour et al. (2020), chitosan coating in combination with whey protein and tarragon essential oil also has the ability to retard the increase in TVN content in *Scomberoides commersonnianus* muscle during storage [9]. In both studies, TVN values correlated well with the microbiological data, indicating that the TVN parameter is a useful index for fish spoilage.

#### 3.2.5. Thiobarbituric Acid-Reactive Substances (TBARS)

Changes in the TBARS values of wrapped and unwrapped herring fillets during storage are presented in Figure 5. TBARS is also considered an indicator of quality for fish exhibiting secondary lipid oxidation products. At the beginning of storage (0 days), the TBARS value was 2.09 mg MDA/kg. The initial TBARS value in our study was slightly lower than that reported at the onset of the refrigerated storage of herring fillets by Tolstorebov et al. (2014) (2.42 mg MDA/kg) [52].

In the unwrapped control sample, secondary lipid oxidation products accumulated, and TBARS values increased to 5.74 mg MDA/kg during 6 days of storage. On the 12th day of storage, a sharp decrease in the TBARS value was recorded, with a further decline during storage. Similar results with increasing and decreasing TBARS values have been recorded in *Scomberoides commersonnianus* during refrigerated storage [9]. Authors have assigned these changes to the partial dehydration of fish and losses of the secondary lipid oxidation products formed during the initial storage period. In our study, this explanation is also reasonable, because the TBARS values only increased and decreased during the storage of the unwrapped herring samples; therefore, they could be dehydrated during storage.

A consistent increase in TBARS during storage was registered for the herring samples wrapped in the different films. However, TBARS values in the herring samples wrapped in composite alginate/pectin film alone were significantly (*p* < 0.05) higher than in those in samples wrapped in films with CE or GE. At the end of the storage period (18 days), TBARS values in the samples coated with film with CE and GE were 2.21 and 2.68 mg MDA/kg, respectively. Higher levels of TBARS were recorded in the samples wrapped in film alone after 18 days of storage (3.37 mg MDA/kg). These results suggest that the supplementation of alginate/pectin films with cranberry or grape seed extracts enhanced the antioxidant properties of the films. The high levels of polyphenols in CE, as well as a large number of procyanidins in GE, act as strong antioxidants that scavenge free radicals and hinder the oxidation chain reactions [53,54]. Cranberry extracts were proven to be suitable for presenting antioxidant (DPPH-scavenging ability) activity in chitosan-based films [27]. Similarly, incorporating grape seed extract into chitosan films improved the oxidative stability of coated red drum fillets [11] and fresh pork [29] under storage with significantly reduced TBARS values.

Notably, all alginate/pectin film-coated samples showed TBARS values of <5 mg MDA/kg throughout the storage period. Meanwhile, the control sample (uncoated) exceeded this value after 4 days of storage. According to Sallam (2007), the maximum TBARS value indicating good fish quality (frozen, chilled or stored with ice) is 5 MDA/kg of tissue [55]. Thus, it can be concluded that according to the values of the lipid oxidation indicator, the quality of the herring remained acceptable during storage when the fish was wrapped in the composite alginate/pectin film supplemented with CE and GE.

## 4. Conclusions

Composite alginate/pectin films supplemented with polyphenol-rich extracts from defatted cranberry pomace (CE) or grape seeds (GE) and applied on herring fillets showed better preservation properties when compared to pure films. The presence of polyphenol-rich extracts considerably suppressed the viability of pathogenic microorganisms (*L. monocytogenes* and *P. aeruginosa*) on the herring fillets. The results indicated a negative influence of films with extracts on the formation of biogenic amines (BAs), which was related to the recorded antibacterial properties of films against bacteria that can be BA producers (e.g., *S. aureus*, *L. monocytogenes*, *S.* Typhimurium and *E. coli*). The application of alginate/pectin films with CE and GE extended the shelf life of herring fillets by minimising pH changes, preventing lipid oxidation and protein degradation and inhibiting microbial growth during storage for 18 days at 4 °C. The results of our study indicate that alginate/pectin films with CE and GE have the potential to be used for the preservation of herring fillets. However, further studies are required to test the influence of these films on the sensory properties of fish.

## Figures and Tables

**Figure 1 foods-12-01678-f001:**
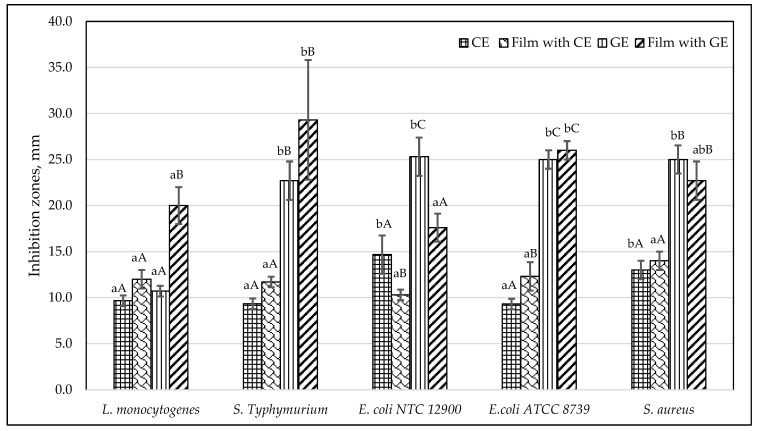
Antimicrobial activities of alginate/pectin films with CE and GE against tested microorganisms. The columns are drawn from the average of at least three independent experiments. Error bars denote ± one standard deviation; small letters indicate significant (*p* < 0.05) differences between bacteria sensitivity to tested alginate/pectin films and extracts; uppercase letters indicate significant (*p* < 0.05) differences between alginate/pectin films and extracts.

**Figure 2 foods-12-01678-f002:**
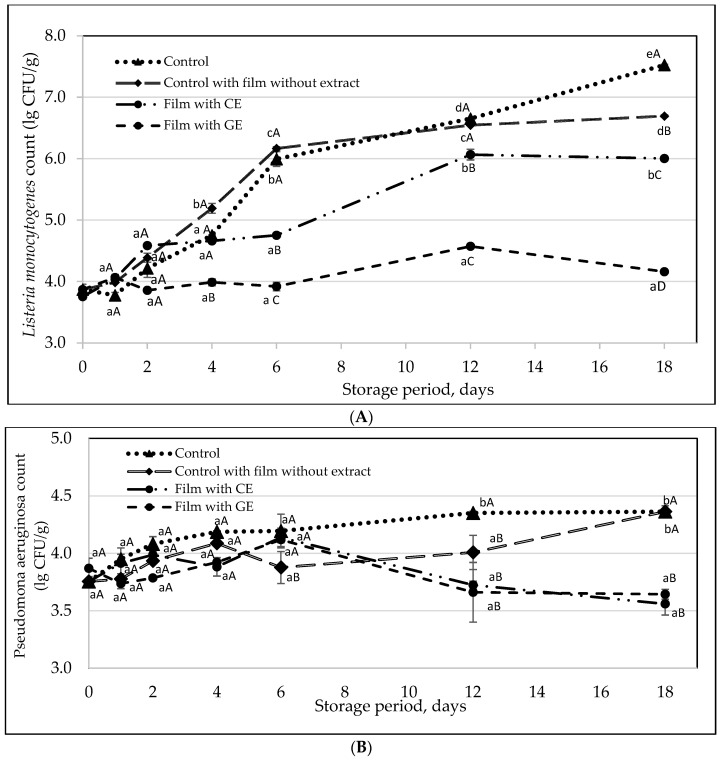
Effect of alginate/pectin film with CE and GE on the growth of *L. monocytogenes* (**A**) and *P. aeruginosa* (**B**) of herring fillets during storage for 18 days. The curves are drawn from the average of at least three independent experiments. Error bars denote ± one standard deviation. Different characters (a, b, c, d, e) indicate significant (*p* < 0.05) differences between storage duration, and A, B, C, D indicate significant (*p* < 0.05) differences between the films. Note: So far as the CFU measurements were not continuous, the lines cannot be used for estimating the precise number of CFU at any storage period between the points.

**Figure 3 foods-12-01678-f003:**
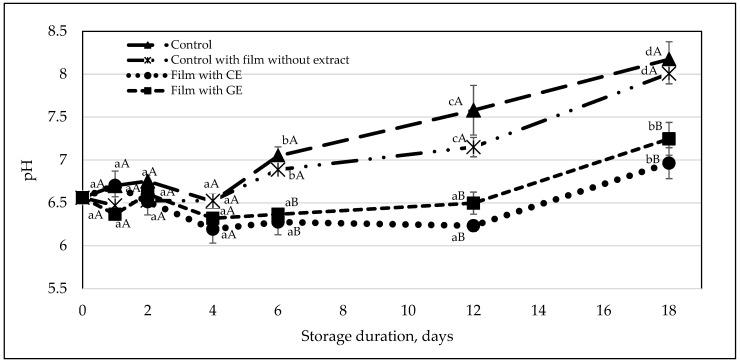
Effect of alginate/pectin film with CE and GE on pH of herring fillets during storage for 18 days. The curves are drawn from the average of at least three independent experiments. Error bars denote ± one standard deviation. Different characters (a, b, c, d) indicate significant (*p* < 0.05) differences between storage duration, and A, B indicate significant (*p* < 0.05) differences between the films.

**Figure 4 foods-12-01678-f004:**
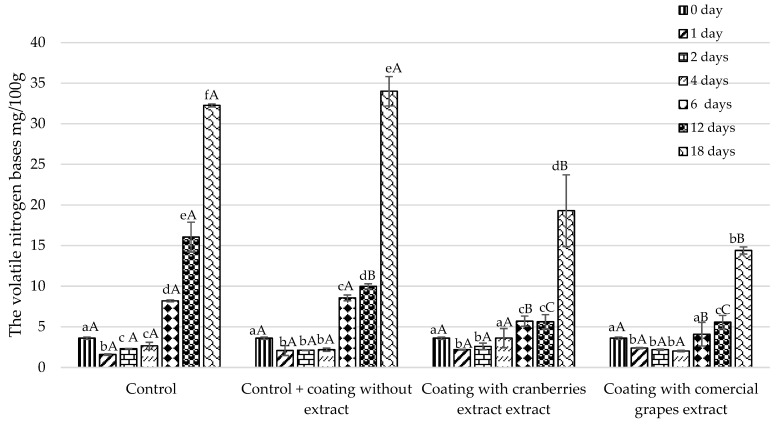
Effect of alginate/pectin film with CE and GE on the total volatile basic nitrogen in herring fillets during storage for 18 days. The columns are drawn from the average of at least three independent experiments. Error bars denote ± one standard deviation. Different characters (a, b, c, d, e, f) indicate significant (*p* < 0.05) differences between storage duration, and A, B, C indicate significant (*p* < 0.05) differences between the films.

**Figure 5 foods-12-01678-f005:**
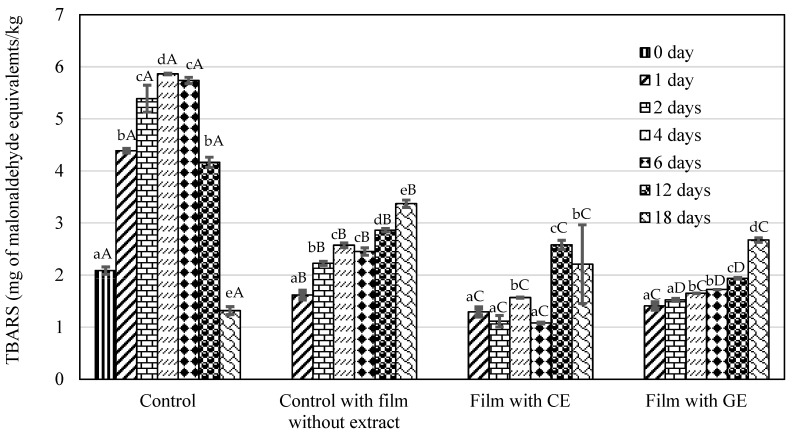
Effect of alginate/pectin film with CE and GE on TBARS in herring fillets during storage for 18 days. The columns are drawn from the average of at least three independent experiments. Error bars denote ± one standard deviation. Different characters (a, b, c, d, e) indicate significant (*p* < 0.05) differences between storage duration, and A, B, C, D indicate significant (*p* < 0.05) differences between the films.

**Table 1 foods-12-01678-t001:** Effect of alginate/pectin film with CE and GE on the changes in biogenic amines (BAs)’ concentrations (mg/kg) of herring fillets during storage for 18 days.

	Sample	Storage Duration, Days					
		1	2	4	6	12	18
HIS	Control	<5 ^aA^	6.11 ± 0.33 ^bA^	<5 ^aA^	<5 ^aA^	58.86 ± 1.53 ^cA^	354.8010.32 ^dA^
	Control with pure film	<5 ^aA^	<5 ^aB^	<5 ^aA^	<5 ^aA^	104.09 ± 2.70 ^bB^	99.35 ± 7.57 ^bB^
	Film with CE	<5 ^aA^	<5 ^aB^	<5 ^aA^	18.67 ± 1.03 ^bB^	38.42 ± 3.42 ^cC^	19.52 ± 2.89 ^bC^
	Film with GE	<5 ^aA^	<5 ^aB^	<5 ^aA^	<5 ^aA^	9.30 ± 0.42 ^bD^	31.11 ± 2.96 ^cD^
CAD	Control	13.02 ± 0.14 ^aA^	17.04 ± 0.2 ^bA^	19.54 ± 0.76 ^bA^	16.30 ± 0.42 ^bA^	139.03 ± 6.32 ^cA^	438.99 ± 9.48 ^dA^
	Control with pure film	<5 ^aB^	<5 ^aB^	<5 ^aB^	15.50 ± 0.71 ^bA^	136.54 ± 5.77 ^cA^	291.32 ± 18.07 ^dB^
	Film with CE	<5 ^aB^	<5 ^aB^	<5 ^aB^	51.20 ± 0.28 ^bB^	26.46 ± 0.91 ^cB^	103.87 ± 6.04 ^dC^
	Film with GE	5.04 ± 0.03 ^aB^	<5 ^aB^	6.23 ± 0.33 ^bC^	<5 ^aC^	26.40 ± 2.97 ^cB^	162.26 ± 4.33 ^dD^
PUT	Control	6.62 ± 0.26 ^aA^	15.53 ± 0.47 ^bA^	<5 ^cA^	14.75 ± 0.64 ^bA^	54.37 ± 3.35 ^dA^	76.81 ± 5.81 ^eA^
	Control with pure film	<5 ^aB^	<5 ^aB^	<5 ^aA^	13.71 ± 1.54 ^bA^	47.73 ± 3.01 ^cB^	118.67 ± 3.07 ^dB^
	Film with CE	<5 ^aB^	6.53 ± 0.75 ^bC^	<5 ^aA^	6.75 ± 0.35 ^bB^	24.30 ± 0.42 ^cC^	15.08 ± 1.53 ^dC^
	Film with GE	<5 ^aB^	<5 ^aB^	<5 ^aA^	8.37 ± 0.35 ^bC^	19.78 ± 1.70 ^cD^	56.65 ± 1.77 ^dD^
TY	Control	<5 ^aA^	5.89 ± 0.30 ^bA^	6.00 ± 0.00 ^bA^	<5 ^aA^	6.08 ± 0.11 ^bA^	25.9 ± 4.24 ^cA^
	Control with pure film	<5 ^aA^	<5 ^aB^	<5 ^aB^	<5 ^aA^	22.10 ± 1.41 ^bB^	49.43 ± 5.98 ^cB^
	Film with CE	<5 ^aA^	<5 ^aB^	<5 ^aB^	5.18 ± 0.26 ^aA^	17.24 ± 2.89 ^bC^	48.02 ± 4.24 ^cB^
	Film with GE	<5 ^aA^	<5 ^aB^	<5 ^aB^	<5 ^aA^	6.71 ± 0.79 ^bA^	6.39 ± 0.25 ^bC^

Different characters (a, b, c, d, e) indicate significant (*p* < 0.05) differences between storage duration, and A, B, C, D indicate significant (*p* < 0.05) differences between the films.

## Data Availability

The data presented in this study are available on request from the corresponding author.

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
