# Peer review of "Alginate/Pectin Film Containing Extracts Isolated from Cranberry Pomace and Grape Seeds for the Preservation of Herring"

_foods, 2023, doi:10.3390/foods12081678_

Round 1

Reviewer 1 Report

Lines 173-174 and throughout the work: Correct the spelling of the Latin name for Salmonella Typhimurium as it is short for: Salmonella enterica sub. enterica ser. Typhimurium. Why were the two strains of Escherichia coli used in the research?

Line 182: Zones of inhibition are usually expressed in [mm] and not in [cm].

Line 265: in the sentence "…. inhibition zones in the range of 1.0–1.4 mm” is inconsistency with the results presented in Fig.1.

Lines 360-365: New "TYR" symbol, not sure what it means?

Reviewer 2 Report

The manuscript by Urbonavičiūtė et al. deal with the development of a new alginate/pectin film containing extracts from cranberry pomace and grape seeds to be use for the extension of shelf life of herring fish.

I suggest major revisions.

The manuscript is interesting. However, some sections are confused, and the line of research is not perfectly clear. Indeed, there is not often correspondence between the order of experiments reported in the material and methods and the results and discussion section, and this make the design of the research not perfectly linear. For example, section 2.3 deal with preparation of herring fillets, section 2.4 deal with antimicrobial test (that do not include fish fillet), and again in section 2.5 experiments with fillet are described.

As the results and discussion are reported together, the manuscript lacks of a real discussion, and this further makes less linear the experimental plan. I suggest the authors to divide results from the discussion.

Beside reported in the material and methods section, statistical analysis is never reported, neither in the text nor in the figure. Indeed, words such as “generally higher”, “greatly increase”, “slightly decrease”, “weak inhibition” have to be replaced with terms such as statistically significant/not significantly different in comparison to the control, along all the manuscript. Also, please add letters showing statistical analysis in the figures.

The authors reported that accordingly with the previous literature, cranberry pomace and grape seeds alone have displayed good antimicrobial properties. Data reported in the current manuscript do not show a great antimicrobial effect of the film (see line 265) and also in the first week of fish storage. Which is the author hypothesis of this effect? Did the authors tested extracts alone (not included as a film) against bacteria used in the current study? Is it possible that the alginate/pectin film does not efficiently release the extract? Which is the extract release rate from the film? Please add this information in the manuscript. Is it possible that film may serve as a nutrient for bacteria, promoting their grow, reducing extract effect? This hypothesis might be tested.

Fish fillets have been inoculated with P. aeruginosa and L. monocytogenes. Which is the microbial load of fish fillets before inoculation?

English has to be revised by a mother tongue. For example, lines 32-34 are not clear.

Please check all abbreviations along all the manuscript. Full name has to be reported the first time in the text, and then always abbreviated. 

 Detailed suggestions:

Line 15: TVN, TBARS with full name, not abbreviated

Line 38: nanoparticles are not from natural sources.

Line 46: A. vera. Full name as not previously reported.

Lines 45- 64: This paragraph is too much long. Please reduce it in length.

Line 76: BA, Full name as not previously reported.

Section 2.1. I suggest to remove this section and include chemicals and brands in the sections where they are used, in order to reduce confusion. Lines 111-120 can be included in a section entitles “extract preparation” or something similar. Lines 121-123 and 125-127 are results and have to be included in the results section. Materials and methods of these data have to be added in the manuscript.

Line 182. Which instrument has been used to measure the inhibition area? A rule? Please note that results are reported with 2 decimal units, that I don’t think is right if you are used a standard rule.

Move section 2.3 after 2.4.

Section 2.5 is not clear. Please rewrite it.

Line 195: what the authors mean as sample? Herring fillets?

Figure 1: Please add the control, i.e. the film without CE and GE.

Line 258: Curves?

Lines 294-295. Please remove them as not necessary. This is a standard methods in the microbiology field.

Figure 2 and 3: What does it mean control? Please add information in the material and methods section. Add statistical analysis. As they are punctiform and not continuous measurements, a connection between points need to be removed.

Lines 304-318, 342-366, 406-422: Describe data taking into account statistical analysis and the comparison with the control.

Lines 422-423: Scomberoides commersonnianus in italic

Round 2

Reviewer 2 Report

The authors only partially revised the manuscript accordingly revision1, and some points still need to be implemented.

The manuscript still lacks of a real discussion, that has not been implemented, neither in part, in any sections.

Beside some concerns were considered by the authors beyond the aim of the manuscript, at least a tentative of discussion could be inserted in the manuscript. For example, the following my comment could be a starting point to implement discussion section.

“The authors reported that accordingly with the previous literature, cranberry pomace and grape seeds alone have displayed good antimicrobial properties. Data reported in the current manuscript do not show a great antimicrobial effect of the film (see line 265) and also in the first week of fish storage. Which is the author hypothesis of this effect? Did the authors tested extracts alone (not included as a film) against bacteria used in the current study? Is it possible that the alginate/pectin film does not efficiently release the extract? Which is the extract release rate from the film? Please add this information in the manuscript.”

Statistical analysis has not been added to figure 2 and 3. Please add it.

The new legend of figure 1 is not clear. Change curves in column.

S. typhymurium always in italic and “typhymurium” in lower case letter along all the manuscript.
